# Measurement of Restrained and Unrestrained Shrinkage of Reinforced Concrete Using Distributed Fibre Optic Sensors

**DOI:** 10.3390/s22239397

**Published:** 2022-12-02

**Authors:** Jacob S. Yager, Neil A. Hoult, Evan C. Bentz, Joshua E. Woods

**Affiliations:** 1Department of Civil Engineering, Queen’s University, 58 University Ave., Kingston, ON K7L 3N6, Canada; 2Department of Civil & Mineral Engineering, University of Toronto, 35 St. George St., Toronto, ON M5S 1A4, Canada

**Keywords:** distributed fibre optic strain sensing, shrinkage, reinforced concrete, restrained shrinkage, unrestrained shrinkage, functionally graded concrete, low carbon concrete

## Abstract

Shrinkage is an important component of the behaviour of reinforced concrete (RC) structures, however, the number of variables that affect shrinkage make it a complex time-dependent phenomenon. Additionally, as new concrete materials with lower embodied carbon gain popularity, there is a need for an in-depth understanding into their shrinkage behaviour before they can be widely adopted by industry. Currently, the shrinkage behaviour of concrete is studied using discrete measurements on small-scale unrestrained prisms. Distributed fibre optic sensing (DFOS) potentially provides a method of measuring both restrained (with reinforcement) and unrestrained (without reinforcement) shrinkage in both small-scale specimens and structural elements. In the current study, methods of measuring distributed unrestrained shrinkage strains were developed and evaluated, and the restrained shrinkage strains in different types of structural members were studied. Unrestrained shrinkage strains were measured using fibres optic cables embedded in small concrete prisms, while restrained shrinkage strains were measured with fibres bonded to the longitudinal reinforcement. Unrestrained shrinkage strains were found to be highly variable (as large as 3800 microstrain range) depending on location, but further research needs to be undertaken to account for end effects, early-stage shrinkage, and bond between the fibre optic cable and the concrete. Restrained shrinkage strains from structural members revealed non-uniform shrinkage strain distributions along member length due to functional grading as well as high supplementary cementitious material concretes, suggesting that shrinkage models will need to account for this variability.

## 1. Introduction

As the use of concrete continues to rise globally and the design of reinforced concrete (RC) structures becomes increasingly more complex, the need for an in-depth understanding of concrete material properties increases. Furthermore, the rise in the use of non-traditional geometries, supplementary cementitious materials (SCMs) [1], and other strategies aimed at reducing the carbon emissions associated with RC has also led to a need for a more detailed understanding of material properties. One such concrete material property is shrinkage, which is a time dependent characteristic of concrete in which a concrete member experiences an overall volume reduction [2]. Strains from shrinkage regularly reach values larger than that of the cracking strain of concrete [3] and since RC is restrained against shrinkage by the presence of reinforcement and adjacent structural components, cracking can occur. Thus, shrinkage is a relevant parameter in reinforced concrete performance in terms of both stiffness and serviceability considerations. Despite its impact on performance, it can be difficult to measure shrinkage in situ and measuring the distribution of shrinkage strain over the length, depth, and width of a member is even more challenging.

Shrinkage is a multifaceted phenomenon that can be divided into two stages and involves several processes [4]. The two stages include: (1) early (including plastic) stage shrinkage, which occurs within 24–48 h after pouring and (2) second stage shrinkage, which occurs after this time. Autogenous shrinkage, drying shrinkage, and thermal effects have the greatest effects over both stages. Additionally, referred to as self-desiccation shrinkage, autogenous shrinkage is the volume reduction due to cement hydration after the initial setting of concrete when no moisture transfer is permitted [5]. Drying shrinkage occurs in the presence of relatively low external air humidity (with respect to the concrete) that causes internal evaporation of water from the cement matrix [6]. Thermal expansive strains also occur as a result of the exothermic cement hydration process creating a steep temperature gradient between inner and outer layers of concrete [4]. The most influential factors on shrinkage include curing conditions (such as ambient relative humidity and presence of water) [7], concrete mix properties (such as water to cement ratio, amount of fine aggregates, presence of replacement binders, and aggregate properties) [8], as well as member geometry [9].

These processes and factors result in complex non-linear three-dimensional behaviour. However, the standard measurement techniques for shrinkage do not reflect its complexity, leaving an incomplete understanding of shrinkage behaviour. One current standardized method to quantify concrete shrinkage is the measurement of change in length of the longitudinal axis of a 285 mm long unreinforced concrete prism [10]. Another commonly used method for measuring combined effects of shrinkage and creep is the stacking of five concrete cylinders in a creep frame, with the shrinkage measured based on the surface shrinkage of the cylinders [11]. With these methods, shrinkage is assumed to be constant and uniform throughout the concrete, and neither method provides a measure of restrained concrete shrinkage, which would be encountered in situ. The single discrete measurement obtained from both methods does not provide any information on how the shrinkage strains are distributed along (or across) a member or how shrinkage is distributed in more complex conditions (e.g., a member with non-prismatic geometry). One promising approach that could shed light on shrinkage distributions, in both restrained and unrestrained conditions, is the use of distributed fibre optic strain sensors (DFOS). 

Rather than providing a discrete strain measurement, DFOS can provide a distributed measurement of strains along the length of a fibre optic cable that is bonded to a substrate material. Other advantages of DFOS include the fact that the cables are relatively inexpensive ($0.15/m), the strain measurement is not affected by electromagnetic interference, and that the fibres are small and corrosion resistant. DFOS have been used extensively to measure strains along reinforcement bars and on concrete surfaces in RC. A Rayleigh backscatter DFOS system is used in this experimental program due to the smaller gauge length (less than 10 mm) and higher accuracy (1 microstrain within the fibre core) [12]. Rayleigh backscatter DFOS has been used for numerous applications in RC. Regier and Hoult [13] were able to utilize Rayleigh backscatter DFOS to detect localised deterioration in rebar in RC. Barrias et al. [14] employed Rayleigh backscatter DFOS to show the feasibility in measuring strains in embedded longitudinal reinforcement as well as to successfully detect cracking. Poldon et al. [15] used Rayleigh backscatter DFOS on longitudinal and transverse reinforcement bars to identify and quantify the strain behaviour of large RC beam shear tests. Yager et al. [16] were able to quantify the longitudinal reinforcement strain behaviour in functionally graded concrete (FGC) beams. It has also been used in multiple scenarios to quantify concrete surface strains [17,18,19].

Several studies have used discrete fibre optic sensors (FOS) to measure unrestrained shrinkage in concrete. Fibre Bragg gratings (FBG) have been used to measure the creep and drying shrinkage of concrete cylinders and showed good agreement with the conventional ASTM C512 method of measuring shrinkage [20]. FBG have also been used to quantify the early age shrinkage of cement paste [21] and unreinforced concrete slabs [22]. Additionally, Fabry–Perot FOS were used to measure unrestrained shrinkage in Hong Kong granite aggregate concrete prisms [23]. Despite the multiple uses of discrete FOS to measure unrestrained shrinkage, the measurement of shrinkage with DFOS has been more limited. 

DFOS has been used in a few instances to get a distributed measure of restrained shrinkage in RC prisms. Davis et al. [24] used DFOS to measure restrained shrinkage along longitudinal reinforcement in the center of 100 × 100 × 900 mm long RC prisms. The study found that the restrained shrinkage profile along the length of the specimen was approximately uniform (±20 microstrain), except at the ends where greater shrinkage occurred (up to −400 microstrain at 28 days compared to −292 microstrain in centre regions). Bado et al. [3] also used DFOS to measure restrained shrinkage in RC prisms that similarly had one longitudinal rebar in the center. These prisms varied in cross section from 80 × 80 mm to 120 × 120 mm and varied in length from 440 mm to 600 mm. The researchers also varied the type of fibre, coating, and fibre placement location and measured both the early stage (not corrected for temperature) and second stage shrinkage. The shrinkage measurement results were found to be in reasonable agreement with values in the literature and initial tensile strains caused by thermal effects were followed by a steady and uniform increase in compressive strains for the second stage shrinkage. Ultimately, the study concluded that the mix design had the largest influence on the shrinkage strain, rather than rebar size. Finally, Poldon et al. [25] used DFOS to calculate the average restrained shrinkage strains in deep beams, however, the study did not report distributed strains.

Despite the advancements in using DFOS to measure shrinkage strains in RC, knowledge gaps remain that can be addressed through the use of DFOS, including distributed unrestrained shrinkage strain behaviour, restrained shrinkage strain in structural members (e.g., beams and slabs), the shrinkage behaviour of low carbon concrete members (high SCM or low cement concrete) and complex functionally graded concrete members. The following experimental campaign explores the use of DFOS as a novel way of measuring unrestrained shrinkage and to characterize distributed restrained shrinkage strain in full-scale and FGC elements for the first time.

The objectives of this research are to: (i) assess the use of DFOS to measure distributed unrestrained shrinkage, (ii) use DFOS to measure restrained shrinkage in large scale structural members and quantify the distributed restrained shrinkage to determine global shrinkage behaviour as well as to characterize local behaviour at locations of interest, (iii) compare values of shrinkage measured using DFOS to existing models and literature, and (iv) compare distributed shrinkage strains from low cement concrete, high SCM concrete, and FGC to distributed shrinkage strains in conventional concrete elements.

## 2. Materials and Methods

### 2.1. Specimens and Instrumentation

The specimens described in this study include those from three experimental campaigns. The first set consists of 15 unrestrained shrinkage prisms measuring 76 × 76 × 285 mm, which is a standard shrinkage specimen size from ASTM C157 [10]. Use of this prism size permitted comparison with conventional shrinkage results taken from literature. The prisms were fabricated using three types of concrete: SCM1, SCM2, and C1. Table 1 shows the 28-day compressive strengths of the three batches of concrete. Mixes SCM1 and SCM2 had at least 30% carbon emission reductions versus ordinary Portland cement concrete according to the supplier, and they were high slag mixes. The C1 concrete was a conventional 35 MPa specified C1 exposure class concrete according to CSA A23.1 [26]. 

Figure 1a shows the fibre optic instrumentation embedded in the prisms. A nylon-coated single mode fibre optic cable was placed at the mid-height of the prism formwork and tensioned to prevent sagging during concrete pouring. The nylon-coated fibre consists of an 8-μm core, 125-μm silica cladding, 250-μm acrylic buffer coating, and a tight and friction fit 900-μm nylon protective jacket. To protect the fibre from damage during demolding, it was placed through a 2.5 mm diameter plastic tube that extended approximately 25 mm into the ends of the specimens and 375 mm out of the specimens. Concrete was placed in the formwork by hand and carefully vibrated (with a poker vibrator) to ensure adequate consolidation. The strain along the length of the fibre was zeroed after vibration (day 0). The prisms were demoulded on day 2 and were placed on rollers, such that all faces were exposed during drying but were not restrained longitudinally. Note that for two of the specimens (SCM1-4 and SCM1-5), the tubing was not embedded far enough into the specimen, which caused a break in the fibre at the end of the specimens during demoulding. That is why there were only 3 specimens of the SCM1 concrete mix.

The second set of specimens were a series of shear critical deep beams (shear span/depth = 2) tested in three-point bending. Additional information on this experimental program can be found in [19]. Figure 1b shows one of the completed beams and the formwork, and Figure 2a shows the geometry of each of the FGC deep beams in the study, which had embedded high strength struts with three different geometries (straight diagonal (DIAG), arch shaped (ARCH), and a diagonal/arch hybrid shape (HYBD)). The beams were constructed with various combinations of low cement concrete surrounding the struts. Table 1 shows the concrete strengths on the day of testing. NSC was the normal strength/normal cement content (425 kg/m^3^) control. Specimens DIAG-LC (low cement) and DIAG-FOH (fresh-on-hardened cast) both contained ultra low cement (150 kg/m^3^) content concrete. Cement savings of up to 47% were achieved compared to NSC for DIAG-LC and DIAG-FOH. The deep beam specimens were poured on their sides as shown in the right image of Figure 1b to create the embedded high strength strut. The FGC beam forms included aluminum dividers used to separate the high strength concrete from the low cement concrete, that were removed after the pouring of each of the layers. For specimens DIAG, DIAG-LC, ARCH and HYBD, the two concretes were poured simultaneously, the dividers were removed, and the concretes were individually vibrated to ensure bond between the two concretes. DIAG-FOH was fresh-on-hardened cast, meaning the strut was cast first, allowed to cure for 24 h, and then the surrounding concrete was poured. After the specimens were demoulded, the beams were placed upright, as shown in Figure 1b. Specimen HYBD was demoulded on day 3, DIAG-FOH on day 4, DIAG, DIAG-LC, and ARCH on day 5, and NSC on day 39.

Figure 1b shows a typical DFOS instrumentation setup for one of the beams (DIAG). The two 20 M bars in each beam were instrumented with fibre optic cables on the top and bottom longitudinal ribs of the bar using the method recommended by Brault and Hoult [27]. The installation process includes: (1) grinding and sanding the longitudinal ribs until smooth, (2) cleaning of the ribs with degreaser, water, and 99% isopropyl alcohol, (3) adhesion of the fibre optic cables to the bars along the ribs using a cyanoacrylate adhesive, and (4) after curing the adhesive for 24 h, a layer of silicone was placed over the fibre optic cables to protect the cables during pouring of concrete. The fibre optic strains were zeroed before pouring of the beams. To account for strains due to temperature change during curing, which are not associated with the shrinkage strains, a thermocouple was attached to each bar in the beams. 

The final specimen-type discussed in this study are a series of uniform and FGC one-way slab strips that were tested in 4-point bending. Figure 1c shows the slab strips and their formwork layout while Figure 2b shows their geometry. The tested slabs included shear critical specimens (designated S), which contained 3–15 M bars and flexure critical specimens (designated F), which contained 2–10 M bars. The concretes used for these slabs were from the same batch as the prisms in the first set of specimens (see Table 1 for strength values). The uniform slab name designations all follow the same format. For example, C-SCM1-F signifies that it is a uniform control (C), contains SCM1 concrete, and is a flexure critical specimen. The FGC slabs were named based on the pouring method. FOH stands for fresh-on-hardened cast (the top layer was poured one week after the bottom layer) and FOF stands for fresh-on-fresh cast (the top layer was poured within 30 min of the bottom layer). The slabs were demoulded at various times after casting: C-SCM1-F, FOH-S, FOH-F, FOF-F, and FOF-S were demoulded on day 3, C-C1-S on day 10, and C-SCM2-F on day 17. 

The fibre optic instrumentation was installed using the same method as the deep beams. Similarly, the fibre optic strains were zeroed just before pouring. The reinforcement bars perpendicular to the longitudinal reinforcement, shown in Figure 2b, were not instrumented with fibre optic sensors. A thermocouple was also installed at mid-height of the slabs to account for temperature changes. 

### 2.2. Test Setup

After casting, all specimens were cured under wet burlap for two days while in the forms. After being removed from their formwork, the specimens were cured in open air indoors as shown in Figure 1. Shrinkage strain measurements were taken using a Luna Technologies ODiSI 6104, which has an accuracy of 1 microstrain within the sensing core and a 2.6 mm gauge length [28]. Measurements were recorded at 1 Hz for 10 s. The strains were then averaged over those 10 s. The internal concrete temperatures were also recorded for the slabs and deep beams and the air temperature next to the concrete was recorded for the prisms. A temperature correction of 20 microstrain/°C was used for the reinforced specimens, which includes an 8 microstrain/°C temperature correction for the fibre [24] and a 12 microstrain/°C correction for the rebar [29]. For the unreinforced specimens, a temperature correction factor of 18 microstrain/°C was applied, which included an 8 microstrain/°C for the fibre and 10 microstrain/°C for the concrete [29]. Various researchers studying applications other than shrinkage have also confirmed the 8 microstrain/°C temperature correction for the analyzer and fibre optic cable combination. Barker et al. [30] confirmed the 8 microstrain/°C temperature correction for fibre optic cables bonded to rail and exposed to 40 °C temperature changes and Mehdi Mirzazadeh and Green [31] found the temperature correction of 8.7 microstrain/°C in a single mode fibre in RC beams under four-point bending and up to 30 °C temperature changes. Strain and temperature measurements in the current campaign were taken periodically until the specimens were tested to failure (those tests are not reported here). 

## 3. Results and Discussion

### 3.1. Measuring Unrestrained Shrinkage in Prisms

Figure 3 shows the 28-day unrestrained shrinkage strain distributions for all the prisms for each of the three concrete mixes. It should be noted that the strains shown in Figure 3 are shown over the 235 mm portion of the fibre that was bonded to the concrete (excluding the protective tubing) and that the results were zeroed on day two (i.e., strains were measured between day 2 and day 28), the reason for which is discussed in detail in following sections. The results in Figure 3 show that while there was significant variability in shrinkage strains at both ends of the prisms, the middle third (80 mm) of the specimens had fairly uniform strains among specimens. Table 2 shows the average unrestrained shrinkage strains over a length of 80 mm in the centre of the prisms as well as the standard deviation amongst the specimens with the same concrete types and the associated coefficient of variation (COV). For specimens fabricated from the same concrete type, the COV of the strain measurements was at most 4.9% at 28 days, signifying limited variability. At all other measurement times, the COV remained at or below 7% (aside from SCM2 at seven days).

Assessing the distribution of shrinkage strains along the prism length, the results in Figure 3 show that the ends of the prisms experienced more variation in the shrinkage strain and in some cases the strains were as high as −100 micrsostrain (compression) and as low as −3700 micrsostrain at 28 days. This was attributed in part to end effects, in which the shrinkage changes due to the ends also being exposed to air. Davis et al. [24] also noted sharp changes in shrinkage strain measurements in prismatic reinforced concrete specimens, however, the strains only varied from −400 micrsostrain to +50 micrsostrain near the ends of the prism, compared to a −292 microstrain average in the centre regions. Another, possible reason for the strain variations at the prism ends was the interaction between the fibre optic cable, concrete, and protective tubing, which could have caused the local compressive strain increase if the fibre was restrained or not straight because of the presence of the tubing. Lastly, development length and bond between the fibre and the concrete could have contributed to the observed localized strains. After the monitoring campaign was complete, it was discovered that the fibres could be pulled out of the specimens with only moderate resistance, suggesting that there was potentially not enough development length to provide reliable strain measurements at the ends of the prisms. 

Another challenge associated with interpreting the shrinkage measurements was the need to tare the data on day two rather than immediately before or after casting. Taring the data means creating a new zero strain reference point. Figure 4 shows the shrinkage strain measurements over the first two days (with respect to a tare that occurred just after pouring the prisms) for the SCM2 specimens. Even in the middle third, which showed approximately constant strain measurements after day 2, the strains vary from −250 micrsostrain to 400 micrsostrain. This variability was most likely the result of early-stage shrinkage processes, including large thermal changes and plastic shrinkage which affected the strain measurements. Ultimately, it was determined that the strains measured over the first 2 days were not representative of the actual behaviour without a more in-depth method to correct for temperature fluctuations, and so, Table 2 only shows second stage shrinkage data (after day 2). 

Figure 5 shows the average shrinkage strains for each type of concrete prism at weekly intervals from day 7 to day 49 and Table 2 summarizes the key results from Figure 5. Figure 5 shows that the shrinkage strain increased rapidly between day 2 and day 14, after which point the shrinkage strain rate began to decrease. By day 35, the shrinkage strains began to level off. Overall, the difference in shrinkage strains between the high SCM concretes and the ordinary C1 mix was minimal (maximum 3.6% difference at 49 days), suggesting that there was little to no difference in the shrinkage behaviour of the three concrete types. The results in Table 2 show that, at 28 days, the shrinkage strains in compression are −485, −490, and −477 microstrain for the SCM1, SCM2, and C1 specimens, respectively. Table 3 shows the shrinkage results taken from the literature for specimens of the same geometry using conventional shrinkage measurement methods. The average of the experimental results from the literature was −258 micrsostrain with a standard deviation of 38.9 micrsostrain, which is about 230 micrsostrain less than the measurements from this experimental campaign. A potential reason for the difference in strains is the inherent variability in shrinkage across different mix designs and curing conditions, as varying material properties and curing conditions can cause significant differences in shrinkage [7]. In one shrinkage measurement campaign [32], the researchers found a COV of 34% for specimens from a single batch of concrete. Thus, concretes with different mix designs and curing conditions would likely produce even larger differences in measured shrinkage strain. Additionally, since the shrinkage strains were tared at day two, thermal tensile strains in the centre of the concrete may have existed at that time, resulting in compressive strains due to concrete cooling in addition to shrinkage strains. These thermal strains were also found by Bado et al. [3] with DFOS. 

Overall, using DFOS, unrestrained shrinkage strains were measured with consistent results in the middle regions (i.e., away from the ends) of small-scale shrinkage test specimens after the first two days of curing. Further research into the nylon-coated fibre bond, end shrinkage, and methods of measuring distributed temperature and unrestrained shrinkage at early ages with DFOS are required to develop a better understanding of unrestrained shrinkage and evaluate the effectiveness of DFOS in measuring unrestrained shrinkage.

### 3.2. Measuring Restrained Shrinkage in Deep Concrete Beams

Another objective of this study was to evaluate the restrained shrinkage behaviour in large-scale RC and FGC members. Figure 6 shows the restrained shrinkage measurements with length and over time, adjusted for temperature (using the method previously described), from one side of a steel reinforcing bar in specimen ARCH. It should be noted that the restrained shrinkage in the concrete is measured indirectly from the strains in the rebar as the specimens exist in a state of self-stress (i.e., the concrete and reinforcing steel are in equilibrium with each other). The strain distributions along the length of the rebar are broken into yellow, blue, and gray zones, which represent distinct shrinkage behaviours. The yellow regions are the locations of the wooden chairs that were used to support the rebar in the form. These chairs are shown in Figure 2b. The local tensile strain peaks in the yellow regions are due to the swelling of the wood due to moisture from the concrete. The strain peaks in the blue region occur at the bend in the rebar. In this region, the concrete shrinks around the bend in the bar resulting in tensile strains on the bottom of the bar due to bending (the fibres on the top of the bar measured compressive strain peaks). Localized shrinkage effects due to inclusions and rebar bends have typically been ignored, but these results highlight that strains in these regions can exceed the cracking strain of the concrete and could potentially cause serviceability concerns.

Figure 6 also shows the locations of the different types of concrete in the beam. The white sections from 500 to 600 mm and 1900 to 2000 mm are regions where the high strength concrete (HF) crosses the rebar, while the grey section is where the low cement concrete (LC) crosses the rebar. The region from 500 to 2000 mm will be the main focus of the restrained shrinkage discussion as it is free from the effects of the rebar bends and wooden chairs. 

The development of shrinkage strains in specimen ARCH over time are also plotted in Figure 6. On day one, there were tensile strains in the LC region. This is once again attributed to the temperature fluctuations during the early-stage shrinkage. Even though internal thermocouples were used to account for temperature strains in these specimens, there likely existed a non-uniform temperature distribution inside the concrete that could not be captured by a single discrete thermocouple measurement and may explain the tensile strain in some regions. The results also show that the shrinkage strains increased from 5 microstrain at midspan to −100 microstrain at midspan over the week after the forms were stripped (between days 5 and 13). This is attributed to the fact that the surface area exposed to drying increased by 177%, indicating that stripping of formwork in large members initiates increased rates of concrete shrinkage. Finally, the results in Figure 6 also show how the distribution of shrinkage strains change throughout the drying process as initially the strain distribution was approximately uniform and changed to a nonlinear distribution with the maximum shrinkage occurring in the centre of the LC region. This phenomenon will be discussed further in the following sections. 

To better understand how the distribution of shrinkage strains varied amongst the test beams with varying geometry (shown in Figure 2a), Figure 7 shows the strain distributions (over the middle 1500 mm region between the wooden chairs) for all beam specimens at or within one day of 28 days. The strain measurements were adjusted to account for temperature as well as localized bar bending by averaging the strain measurements on two sides of the same bar, as described by Davis et al. [24]. The results in Figure 7 show large variations in the shrinkage strain distributions within each beam, which is attributed to the concrete mix, and more specifically the amount of cement in the mix. Past work by Bentz and Snyder [39] has shown that cement content is a primary factor that influences shrinkage behaviour of concrete. The NSC specimen, which was made of Ordinary Portland Cement concrete (OC) had an approximately uniform shrinkage strain distribution of −148 microstrain on average. The local peaks in the strain distribution could be attributed to the coarse aggregate in the concrete contacting the fibre optic cable or air pockets, causing a local strain disturbance. Beams DIAG, ARCH, and HYBD were all fabricated of concrete with the same mix designs and distribution of concrete at the rebar location: the outer 100 mm on each end of each FGC specimen was high strength (HF) concrete, and the inner 1300 mm was low cement (LC) concrete. The results show that these specimens exhibited similar magnitude and distribution of shrinkage strains, which had a distinct curved distribution. Finally, beams DIAG-FOH and DIAG-LC were both fabricated of ultra low cement concrete (ULCC), which had a much lower cement content, and the results show that both of these specimens had much smaller shrinkage strains.

Figure 8 shows the evolution of the shrinkage strain distribution over time for NSC, DIAG-FOH, ARCH, and HYBD. The results show that local peaks in the strain distributions were present in the initial readings after early-stage shrinkage and persisted in subsequent strain measurements. The strain distribution for specimen NSC (Figure 8a) was approximately uniform and increased uniformly from 18 to 28 days and had an average shrinkage strain of −148 microstrain with a standard deviation of 12.8 microstrain. Specimen Diag-FOH (Figure 8b) also exhibited a relatively uniform shrinkage strain distribution, however, the magnitude of the shrinkage strains was very low and in some cases in tension, which is attributed the low cement content of the concrete. Exacerbating the effects of the low cement content was significant leakage of bleed water mixed with cement during concrete pouring which resulted in even lower cement contents, low water to cement ratios, and extremely low shrinkage of the concrete. In the remaining FGC beam specimens (ARCH and HYBD in Figure 8c,d), the results show that while the shrinkage strain distribution was initially uniform (at 7 and 8 days), there was a non-uniform increase in shrinkage strains along the member length as a larger increase in strain was observed in the middle region of the specimen, creating a curved strain distribution. Because the strain distribution was time dependent, this behaviour is attributed to the fact that these beams were functionally graded and that the different concretes used within the member experienced different time-dependent shrinkage behaviours leading to non-uniform shrinkage strain distributions along the beam length. In particular, the presence of an embedded strut (shown in Figure 2) had a tendency to prevent shrinkage near supports while the middle of the beam experienced higher shrinkage, leading to the nonlinear shrinkage strain distributions observed in Figure 8. 

This observed behaviour is significant because it indicates that in concrete members with complex geometries or different concrete properties, this can lead to complex shrinkage behaviour and could result in larger than expected shrinkage strains. For example, the FGC deep beams examined in this study experienced shrinkage strains that were up to 100 micrsostrain larger than the NSC beam. These larger strains could result in more significant shrinkage cracks that can have a negative impact on the overall performance of a structure. Ultimately, these results show that restrained shrinkage behaviour in large-scale structural members, particularly those with unique geometry or functional grading can be complex, and the use distributed strain measurements from DFOS can shed light on this behaviour. 

### 3.3. Measuring Restrained Shrinkage in One Way Concrete Slabs

To understand the distribution of shrinkage strains in members with more conventional geometry, Figure 9 shows the 28-day shrinkage strain measurements along the length of one steel reinforcing bar (including compensation for temperature and bar bending) in each of the one-way slab strip specimens. The results show local peaks in the strain distributions, which are once again attributed to the interaction between aggregates or air pockets in the concrete and the fibre optic cable. At approximately 1500 mm, the large strain peaks in tension were caused by the presence of the plastic chairs used to support the steel reinforcement, which could have pinched the fibre cable at that location.

The results in Figure 9 shows that for the normal concrete control (C-C1-S), the shrinkage strain distribution was approximately uniform along the length of the bar, with an average shrinkage strain of −157 micrsostrain and a standard deviation of 15.0 micrsostrain. The results also show that all the specimens that had bars surrounded by high SCM concrete (FOF-F, FOF-S, FOH-F, FOH-S, C-SCM1-F, and C-SCM2-F), the shrinkage strain distribution was nonlinear and upward curving, opposite to those observed in the deep beam specimens. Furthermore, the bars that were surrounded by SCM1 concrete all had similar strain distributions and comparable values of average shrinkage strain. This suggests the horizontal layering of the FGC specimens as well as the different rebar sizes (15 M for “S” specimens and 10 M for “F” specimens) had little effect on the restrained shrinkage behaviour of these specimens. 

Figure 10 shows the development of strains for C-SCM1-F (Figure 10a,c) and C-SCM2-F (Figure 10b,d) for a tare prior to the pour and for a tare after day 2. The results show that the nonlinear distribution forms during the first 2 days of curing, and the strain distributions with a tare at day 2 are approximately uniform for both specimens. This suggests that the nonlinear strain distribution was once again caused by early-age shrinkage, including thermal and plastic shrinkage strains that initiate in the first two days, with the strain distributions remaining relatively locked in thereafter. This differs from the deep beam specimens in which the FGC beams nonlinear strain distribution developed over weeks instead of days. Furthermore, Figure 10a,c show that different bars in the same beam had similar strains, suggesting that the shrinkage strains remained relatively constant through the width of the beam. 

Figure 10c also shows a local strain peak and valley in specimen C-SCM2-F at 2400 mm, the shape of which somewhat resembles a trigonometric tangent curve. This localised strain behaviour has been previously found during testing of beam specimens by Poldoen et al. [15]. However, these had not been recorded during shrinkage strain measurements until this test. The results in Figure 10c show the progression of the distributed strain trigonometric tangent curve (DSTTC) in C-SCM2-F from day 2 to day 28. Figure 10d, which shows the day two tare of the same specimen, does not have a DSTTC. This indicates that the formation of the DSTTC occurred before day two, and the magnitude of the DSTTC did not change significantly over time. After the specimen was tested to failure (not described in this study), the concrete surrounding the region of the bar in the vicinity of the DSTTC was chipped away to look for a possible reason for why the DSTTC occurred in that region. An air pocket was found in the location of the DSTTC, shown in Figure 11, between two transverse ribs (~5 mm length). While an air pocket was associated with the DSTTC in this experimental campaign, the phenomena of the DSTTC as a general behaviour requires further research. This monitoring of this phenomenon, nevertheless, demonstrates the ability of DFOS to locate imperfections, such as air pockets, before loading occurs.

Another interesting parameter that can be investigated using the DFOS data collected in this study is the influence that stripping of the formwork had on the magnitude and distribution of the shrinkage strains along the slab strip specimens. FOF-F, FOF-S, FOH-F, FOH-S, and C-SCM1-F were demoulded three days after pouring, while specimens C-C1-S and C-SCM2-F were demoulded 10 days and 17 days after pouring, respectively. Figure 12 shows the second stage (tare at day two) restrained shrinkage averaged between 500 and 1000 mm for one bar in each of the three uniform specimens. The time of demoulding for each specimen is shown with a colour coded vertical line. The results show that following demoulding, there are rapid increases in the shrinkage strains. Furthermore, the earlier the demoulding, the larger increase in shrinkage strain at 28 days. This is attributed to the fact that stripping the formwork earlier exposes a larger surface of the concrete to drying for longer, compared to having a single face exposed to drying while in the forms. Time of demoulding has also been found by Samouh et al. [40] to play a role in the magnitude of shrinkage strains.

### 3.4. Comparison of DFOS Shrinkage Measurements to Code Procedures

To evaluate shrinkage strains from DFOS, a comparison was made with the ACI 209R-92 model [41]. This model was chosen as it enables shrinkage strain estimation without the need for a wide range of input parameters (e.g., curing and ambient conditions, mix design) and recommends typical values. While having more detailed input parameters would make the model more accurate, an in-depth set of parameters was not available for the specimens presented in this paper, and so, typical values were used. 

Table 4 shows the shrinkage strain predictions from ACI 209R-92 model for the unrestrained prisms and restrained uniform slabs and beams at 28 days as well as the measured average shrinkage strains for the prisms, slabs, and beams. Because wet burlap was removed from the concrete on day 2, the start of drying was also considered to be on day 2. It is noted that because the effects of functional grading cannot be quantified by the ACI model, predictions of the shrinkage strain in the functionally graded beams and slabs shrinkages were not carried out.

Each of the prisms had a predicted unrestrained shrinkage strain of 343 microstrain since nothing other than mix design differed amongst the specimens. Table 4 shows the shrinkage strain results for the prisms, which were similar in magnitude, however, SCM1, SCM2, and C1 had unrestrained shrinkage values that were 41%, 43%, and 39% greater than the ACI prediction. This difference is relatively typical for typical shrinkage measurements compared to models [42]. Al-Saleh [42] saw measured shrinkage strains of up to 174% greater than the ACI model at high temperatures, but typical differences at room temperature were about 50% compared to the ACI model. 

The percentage difference between the experimental data and the ACI model is even lower for many of the restrained specimens. The restrained ACI model prediction was calculated by determining the unrestrained shrinkage (considering varying volume to surface area ratios and demoulding times), and then translated to the restrained shrinkage strain through a compatibility and equilibrium analysis of the section. Table 4 shows the differences in restrained shrinkage for the beams and slabs compared to the ACI model, which were 44% smaller for C-SCM1-F, 0.5% smaller for C-SCM2-F, 18% larger for C-C1-S, and 111% larger for NSC. The results show that the prediction from the ACI model, despite having limited available input parameters, fared relatively well for some specimens. However, for NSC, the measured results varied significantly from the prediction, and even among the other specimens, there was significant variability in the percentage differences. It is well known that the shrinkage behaviour of concrete can be hard to predict, and these results suggest that simplified models may not be the most reliable approach to estimate shrinkage. 

However, results in this study show that DFOS can provide valuable insights into the distribution of restrained shrinkage strains throughout RC members. In particular, results have shown that FGC members or those fabricated using SCM concrete have non-uniform shrinkage strain distributions that vary by over 100 microstrain along the length of the member. However, within ordinary concrete beams and slabs, that have reasonably uniform shrinkage strains, DFOS allows for measurement of shrinkage strains every 2.6 mm, which can be used to create a distribution of shrinkage strain, something that would not be possible with conventional discrete sensing methods. Figure 13 shows the distributions of shrinkage strains for the ordinary concrete members in this study. In both ordinary concrete specimens, while extreme strain variations exceeded 45 microstrain, the majority of measurements fall near the average, and the results show that the shrinkage strains are approximately normally distributed along the member length.

## 4. Conclusions

This study investigated the use of DFOS to measure unrestrained and restrained shrinkage strains in reinforced, functionally graded, and low cement concretes. Unrestrained shrinkage was measured in small-scale prisms while restrained shrinkage was measured along the longitudinal steel reinforcement in deep beams and one-way slab strips. The following are the key findings of the study:DFOS can be used to measure the distributed unrestrained shrinkage strain profiles along the length of a concrete prism. However, the results shed light on the complexity of unrestrained shrinkage in the test specimens, in terms of both its spatial and temporal variation, suggesting that further research is required to better understand the end behaviour, fibre bond effects, early-stage unrestrained shrinkage, and thermal effects. The current experimental campaign only reported second stage shrinkage strains and averaged strains from the middle third to account for these complexities.DFOS were able to measure the restrained shrinkage behaviour in large-scale deep beams and one-way slab strips. In concrete members with prismatic geometry fabricated with Ordinary Portland Cement concrete, restrained shrinkage strains were uniformly distributed and were −153 microstrain at 28 days on average for the studied specimens. A normal distribution of shrinkage strain measurements existed in these specimens.Distributed restrained shrinkage strain measurements in functionally graded deep beams with embedded high strength concrete struts had nonlinear shrinkage strain profiles that had maximum compressive strains in the centre of the beam and minimum strains at the ends. These distributions formed over time rather than during early-stage shrinkage. However, in slab strips with horizontal functional grading above the reinforcement, functional grading was shown to have a minimal effect on the shrinkage behaviour. Overall, functional grading only affected restrained shrinkage strain distributions if the grading crossed the reinforcement.Distributed restrained shrinkage strain measurements were also found to vary depending on the concrete mix. While the uniform normal concrete had uniform shrinkage strain distributions, the ultra low cement concretes without SCMs experienced shrinkage strains that were up to 80% lower compared to the normal concrete and were less influenced by the presence of functional grading. Low cement concretes with SCMs had nonlinear strain distributions that formed during early-stage shrinkage, and only uniform increases in shrinkage occurred thereafter.Compared to other researchers, the unrestrained shrinkage strains were approximately 200 microstrain higher, which is attributed to the inherent variance of shrinkage measurements and thermal effects. Furthermore, the unrestrained and restrained shrinkage strains for uniform specimens were compared to the ACI 209 model. The ACI model, while providing relatively accurate predictions for some specimens, did not capture the variability that exists with concrete shrinkage.

## Figures and Tables

**Figure 1 sensors-22-09397-f001:**
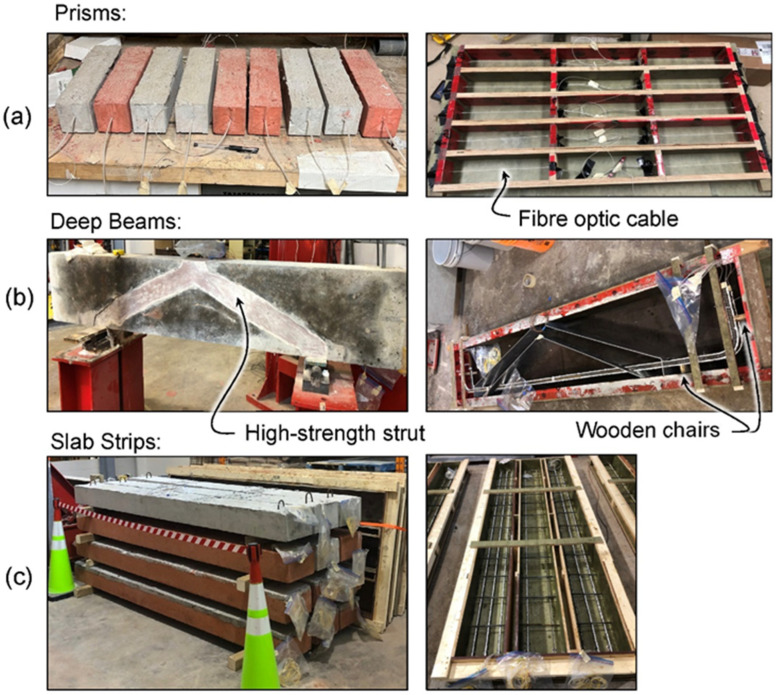
Specimens in curing conditions (**left**) and DFOS instrumentation pre-pour (**right**) (**a**) Prisms, (**b**) Deep Beams, (**c**) Slabs.

**Figure 2 sensors-22-09397-f002:**
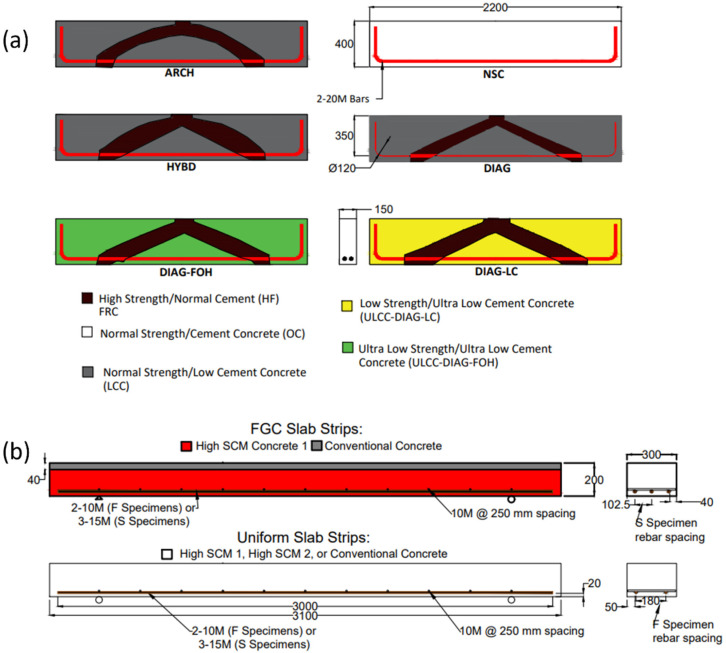
Specimen dimensions and functional grading—all dimensions in mm (**a**) Deep Beams, (**b**) Slabs.

**Figure 3 sensors-22-09397-f003:**
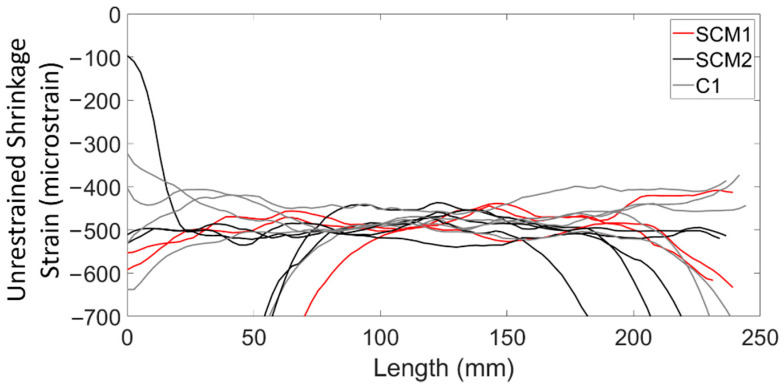
Unrestrained shrinkage strain of prisms at age of 28 days.

**Figure 4 sensors-22-09397-f004:**
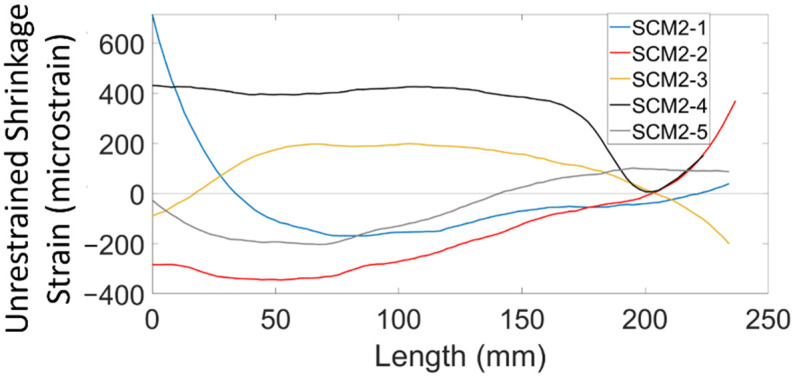
Unrestrained shrinkage strain of SCM2 prisms at an age of 2 days.

**Figure 5 sensors-22-09397-f005:**
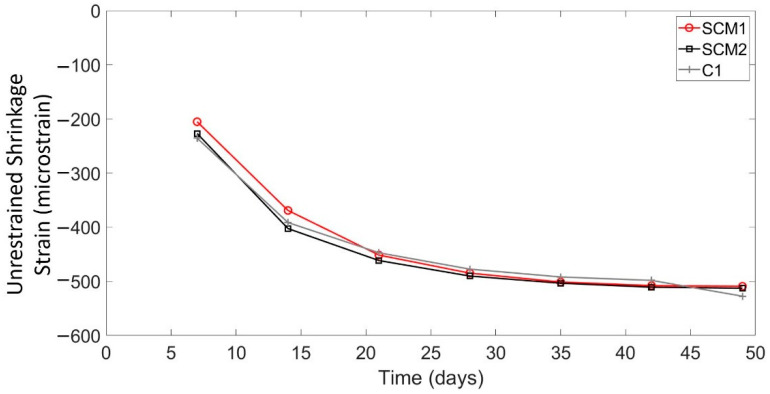
Unrestrained shrinkage strain over time.

**Figure 6 sensors-22-09397-f006:**
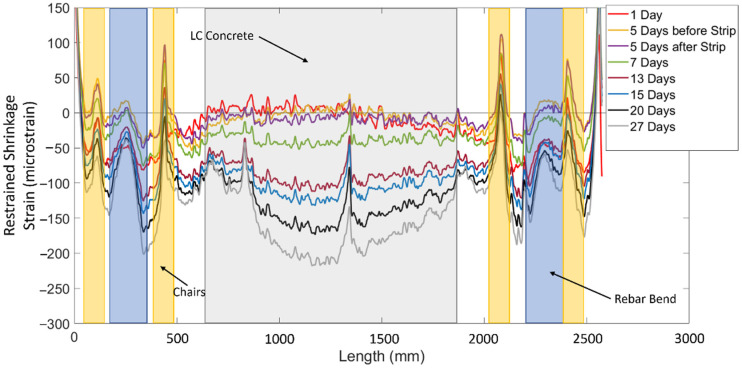
Shrinkage strain with length for specimen ARCH at different ages—one side of bar.

**Figure 7 sensors-22-09397-f007:**
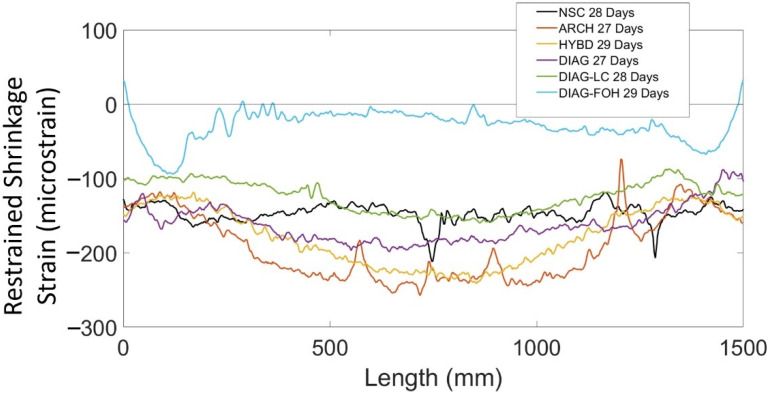
Shrinkage of all deep beams over the middle 1500 mm of the span at 28 days.

**Figure 8 sensors-22-09397-f008:**
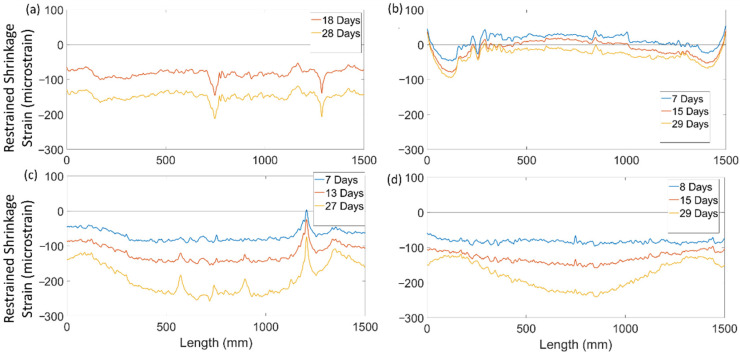
Shrinkage of deep beams at different ages—averages of both sides of bar in specimen (**a**) NSC, (**b**) DIAG-FOH, (**c**) ARCH, and (**d**) HYBD.

**Figure 9 sensors-22-09397-f009:**
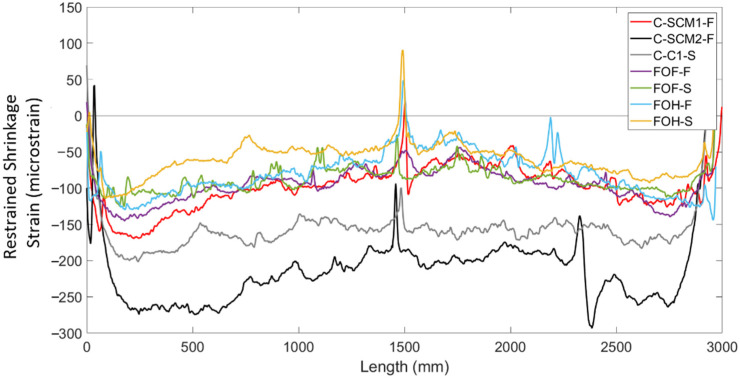
Shrinkage of all slabs at 28 Days.

**Figure 10 sensors-22-09397-f010:**
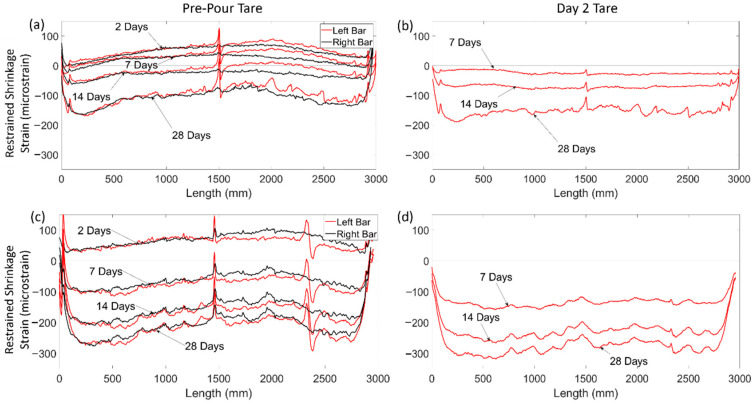
Shrinkage of uniform SCM slabs at different ages (**a**) C-SCM1-F pre-pour tare, (**b**) C-SCM1-F 2-day tare, (**c**) C-SCM2-F pre-pour tare, and (**d**) C-SCM2-F 2-day tare.

**Figure 11 sensors-22-09397-f011:**
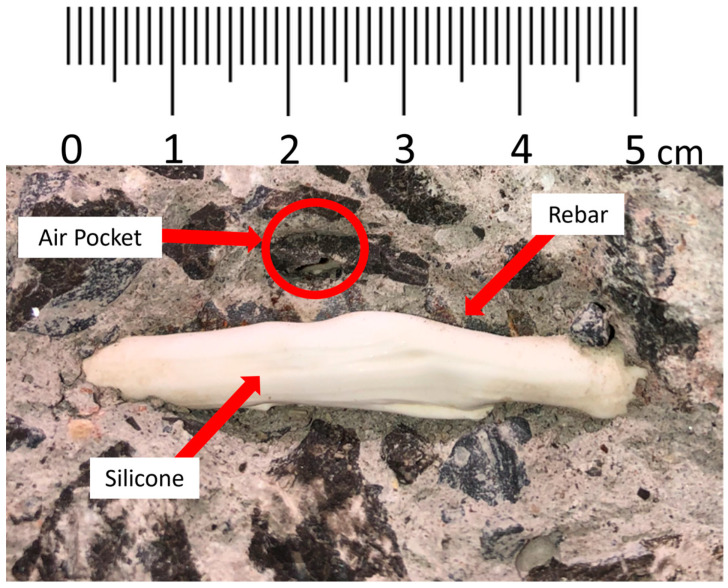
Air pocket found in C-SCM2-F.

**Figure 12 sensors-22-09397-f012:**
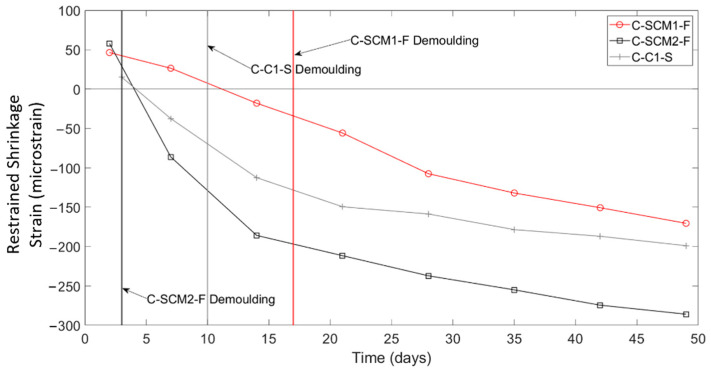
Average shrinkage strain in uniform concrete slabs over time.

**Figure 13 sensors-22-09397-f013:**
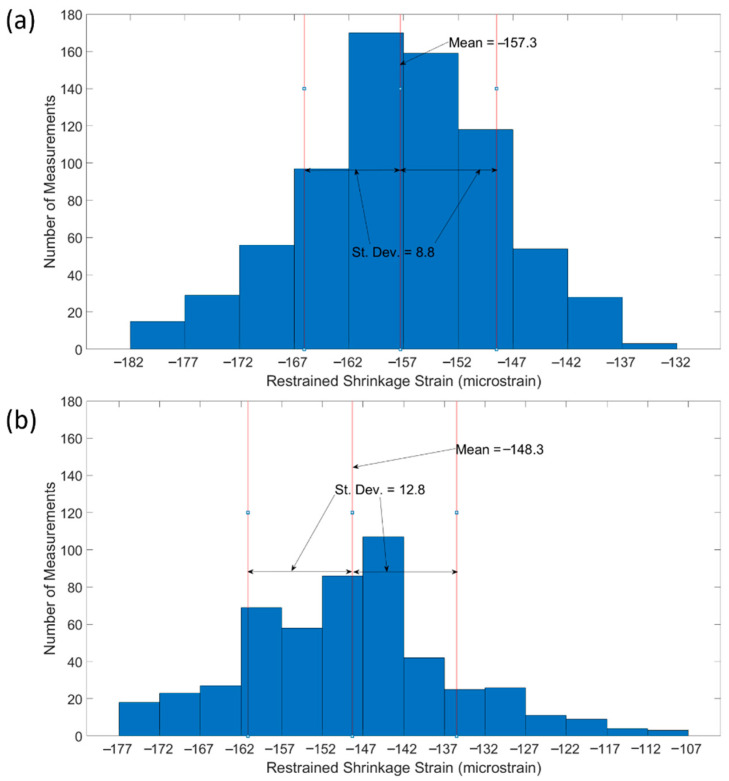
Restrained shrinkage strain distributions in ordinary concrete specimens (**a**) C-C1-S, (**b**) NSC.

**Table 1 sensors-22-09397-t001:** Concrete compressive strengths and age at testing.

Concrete Type	Average Compressive Strength (MPa)	Age at Test (Days)
SCM1	24.6	28
SCM2	24.4	28
C1	40.8	28
OC-NSC	49.7	118
LCC-ARCH	42.9	49
LCC-HYBD	45.6	46
LCC-DIAG	47.0	49
ULCC-DIAG-LC	24.8	28
ULCC-DIAG-FOH	11.3	29
HF-ARCH	69.6	49
HF-HYBD	73.5	46
HF-DIAG	69.7	49
HF-DIAG-LC	76.1	28
HF-DIAG-FOH	79.8	28

**Table 2 sensors-22-09397-t002:** Average unrestrained shrinkage over time (microstrain).

Day	7	14	21	28	35	42	49
SCM1 Average	−205	−369	−451	−485	−501	−508	−509
SCM1 St. Dev.	8.2	24.3	17.2	20.2	22.6	23.8	25.2
SCM1 COV	4.0%	6.6%	3.8%	4.2%	4.5%	4.7%	4.9%
SCM2 Average	−227	−403	−462	−490	−503	−511	−513
SCM2 St. Dev.	42.6	21.5	18.3	22.1	28.1	32.3	35.6
SCM2 COV	18.8%	5.3%	4.0%	4.5%	5.6%	6.3%	7.0%
C1 Average	−236	−391	−447	−477	−492	−498	−527
C1 St. Dev.	8.3	16.7	20.3	23.5	25.4	26.9	27.9
C1 COV	3.5%	4.3%	4.5%	4.9%	5.2%	5.4%	5.3%

**Table 3 sensors-22-09397-t003:** Unrestrained shrinkage measurements from literature.

Experimental Comparison	Shrinkage Strains at 28 Days (Microstrain)
[33]	−200
[34]	−290
[35]	−310
[36]	−220
[37]	−250
[38]	−280
Average	−258
St. Dev.	38.9

**Table 4 sensors-22-09397-t004:** Average shrinkage measurements compared to ACI model.

Concrete Type	Prism Unrestrained Shrinkage at 28 Days (Microstrain)	ACI Model Prediction (Microstrain)	% Difference	Slab/Beam Unrestrained Shrinkage at 28 Days (Microstrain)	ACI Model Prediction (Microstrain)	% Difference
SCM1	485	343	+41%	55	99	−44%
SCM2	490	343	+43%	212	213	−0.5%
C1	477	343	+39%	157	133	+18%
NSC	-	-	-	148	70	+111%

## Data Availability

Not applicable.

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
