# Peer review of "Measurement of Restrained and Unrestrained Shrinkage of Reinforced Concrete Using Distributed Fibre Optic Sensors"

_sensors, 2022, doi:10.3390/s22239397_

Round 1

Reviewer 1 Report

The articles describe the measurement of restrained and unrestrained shrinkage of reinforced concrete using distributed fiber optic sensors. Although the authors have tried to cover all the aspects and explain them, the following points need to consider for the possible publication of the proposed work. 

1. The manuscript needs extensive English editing and sentence restructuring.

2. Abstract should reflect the background, objectives, methodology, and results, including the novelty of the work.

3. The article does not reflect the actual novelty of the work. 

4. If only the strain is measured through the fiber optic sensors, then what is the work's novelty? Highlight it. 

5. The text is not aligned properly, and the template is missing for the manuscript. 

6.  Keywords are always below the abstract. 

7. Figures are not indicative.

Author Response

Response to reviewers for Sensors-2075213

Reviewer 1:

Comment

Response

1. The manuscript needs extensive English editing and sentence restructuring.

The manuscript has been carefully checked and proofread by four native English speakers.

2. Abstract should reflect the background, objectives, methodology, and results, including the novelty of the work.

The authors have followed this format in the creation of the abstract.

3. The article does not reflect the actual novelty of the work. 

Details of the novelty has been expanded upon in the Introduction: “The following experimental campaign explores the use of DFOS as a novel way of measuring unrestrained shrinkage and to characterize restrained shrinkage strain in full-scale and FGC elements for the first time.”

4. If only the strain is measured through the fiber optic sensors, then what is the work's novelty? Highlight it. 

See above response where the new text has been added to highlight the novelty.

5. The text is not aligned properly, and the template is missing for the manuscript. 

The manuscript has been edited to use the template of the journal.

6.  Keywords are always below the abstract. 

The manuscript has been edited to use the template of the journal, which has the keywords below the abstract.

7. Figures are not indicative.

The authors were not able to address this comment without more specific detail as we feel that the figures are indicative of the results and conclusions presented in the paper.

Reviewer 2 Report

The paper presents 2 type of results. One type concerns to several facts an approaches related to the shrinkage of concrete as a main property of the material. The other part is related to the ability of DFOS to measure the strains along the rebar to obtain the profile of strain in the rebar due to shrinkage. It is clear that this second aspect, which is the most interesting subject for the readers of sensors, has been clearly demonstrated by the tests explained in the present paper as well as other papers on the subject that are properly referenced. It is also recognized that the use of optical fiber with low bonding to concrete is not a correct way of measuring unrestrained shrinkage. However, although not being the main subject of the present paper, there are some concerns about some results that this reviewer would like to raise. 

1.- In the case of unrestrained shringkage, the authors fully recognize the differences between their results and those normally seen in the available literature and guidelines or codes on concrete properties (in fact the last conclusion states: "Compared to other researchers, the unrestrained shrinkage strains were approximately 200 microstrain higher, which is attributed to the inherent variance of shrinkage measurements ". I would like that the authors develop more on this difference and argue whether the high difference could be the result of "inherent variance of shrinkage measurements" or because they start the measurements after day 2 of the pouring, where the concrete has been subject to thermal expansive strains. Therefore, when starting the measurement in day 2, they not only measure the shrinkage of concrete but also the decrease in strain due to the relaxation of the temperature effects occuring from pouring until day 2. By the way, the authors should also clarify what they mean by "inherent variance of shrinkage measurements"

2.- In relation with the previous comment and concerning the restrained shrinkage, it should be noted that the shrinkage in the concrete is indirectly measured by the strain in the rebars. Again, this is why they measure tension in the rebars up to the second day and taring at this day provide realistic results.

Some minor comments:

1.- In page 10: ".......and Table 2 shows summarizes the key results......" shows or summarizes should be removed

2.- In page 13: Something is missing in the phrase: "In beams DIAG, ARCH, and HYBD, which were all fabricated of concrete with the same mix designs and distribution of concrete"

Author Response

Response to reviewers for Sensors-2075213

Reviewer 2:

Comment

Response

1.- In the case of unrestrained shringkage, the authors fully recognize the differences between their results and those normally seen in the available literature and guidelines or codes on concrete properties (in fact the last conclusion states: "Compared to other researchers, the unrestrained shrinkage strains were approximately 200 microstrain higher, which is attributed to the inherent variance of shrinkage measurements ". I would like that the authors develop more on this difference and argue whether the high difference could be the result of "inherent variance of shrinkage measurements" or because they start the measurements after day 2 of the pouring, where the concrete has been subject to thermal expansive strains. Therefore, when starting the measurement in day 2, they not only measure the shrinkage of concrete but also the decrease in strain due to the relaxation of the temperature effects occuring from pouring until day 2. By the way, the authors should also clarify what they mean by "inherent variance of shrinkage measurements"

The authors thank the reviewer for this insight. The authors have added discussion to address the reviewer’s points: “A potential reason for the difference in strains is the inherent variability in shrinkage across different mix designs and curing conditions, as varying material properties and curing conditions can cause significant differences in shrinkage [7]. In one shrinkage measurement campaign [38], the researchers found coefficient of variations (COV) of 34% for specimens from a single batch of concrete. Thus, concretes with different mix designs and drying conditions would likely produce even larger differences in measured shrinkage strain. Additionally, since the shrinkage strains were tared at day two, thermal tensile strains in the centre of the concrete may have existed at that time, resulting in compressive strains due to concrete cooling in addition to shrinkage strains."

2.- In relation with the previous comment and concerning the restrained shrinkage, it should be noted that the shrinkage in the concrete is indirectly measured by the strain in the rebars. Again, this is why they measure tension in the rebars up to the second day and taring at this day provide realistic results.

The authors have included in the manuscript that the restrained shrinkage measurements are indirectly measured: “It should be noted that the restrained shrinkage in the concrete is measured indirectly from the strains in the rebar as the specimens exist in a state of self-stress (i.e. the concrete and reinforcing steel are in equilibrium with each other).”

1.- In page 10: ".......and Table 2 shows summarizes the key results......" shows or summarizes should be removed

The authors thank the reviewers for catching these grammar mistakes. This has been fixed.

2.- In page 13: Something is missing in the phrase: "In beams DIAG, ARCH, and HYBD, which were all fabricated of concrete with the same mix designs and distribution of concrete"

This sentence has been fixed it now reads: “Beams DIAG, ARCH, and HYBD were all fabricated of concrete with the same mix designs and distribution of concrete at the rebar location: the outer 100 mm on each end of each FGC specimen was high strength (HF) concrete, and the inner 1300 mm was low cement (LC) concrete.”

Reviewer 3 Report

The comments on the manuscript entitled "Measurement of restrained and unrestrained shrinkage of reinforced concrete using distributed fibre optic sensors" by Yager et al.:

1. The manuscript presents the use of distributed fiber optic sensing for measuring unrestrained and restrained shrinkage strains in reinforced, functionally graded, and low cement concretes. The manuscript is a bit long and and could be shortened to improve its readability and efficiency.

2. The references should be cited as journal format.

3. please mention the parameters of the nylon-coated single mode fiber optic cable which can be useful for readers.

4. Is there any environmental study on the distributed fiber optic sensors performance such as temperature? Please clarify.

5. It is suggested to compare the results with previous reports.

Author Response

Response to reviewers for Sensors-2075213

Reviewer 3:

Comment

Response

1. The manuscript presents the use of distributed fiber optic sensing for measuring unrestrained and restrained shrinkage strains in reinforced, functionally graded, and low cement concretes. The manuscript is a bit long and and could be shortened to improve its readability and efficiency.

The authors thank the reviewer for suggestions on how to improve the readability. The manuscript has been further edited and extraneous sentences have been removed to improve readability

2. The references should be cited as journal format.

The authors have fixed the reference format.

3. please mention the parameters of the nylon-coated single mode fiber optic cable which can be useful for readers.

The authors thank the reviewer for requesting this information that would be quite relevant to Sensors readers. The authors added information on the cable to: “The nylon-coated fibre consists of an 8-μm core, 125-μm silica cladding, 250-μm acrylic buffer coating, and a tight and friction fit 900-μm nylon protective jacket.”

4. Is there any environmental study on the distributed fiber optic sensors performance such as temperature? Please clarify.

In addition to the thermal calibration provided by [24], the authors have also added information from other studies that dealt with DFOS and their behaviour in changing temperatures: “Various researchers studying applications other than shrinkage have also confirmed the 8 microstrain/°C temperature correction for the analyzer and fibre optic cable combination. Barker et al. [30] confirmed the 8 microstrain/°C temperature correction for fibre optic cables bonded to rail and exposed to 40 °C temperature changes and Mehdi Mirzazadeh and Green [31] found the temperature correction of 8.7 microstrain/°C in a single mode fibre in RC beams under four-point bending and up to 30 °C temperature changes.”

5. It is suggested to compare the results with previous reports.

The authors agree that comparing results to previous reports is important for this manuscript. The following text addresses this comment: The authors have compared the shrinkage results with unrestrained shrinkage results from literature (Table 3): “Table 3 shows the shrinkage results taken from the literature for specimens of the same geometry using conventional shrinkage measurement methods. The average of the experimental results from the literature was -258 micrsostrain with a standard deviation of 38.9 micrsostrain, which is about 230 micrsostrain less than the measurements from this experimental campaign. A potential reason for the difference in strains is the inherent variability in shrinkage across different mix designs and curing conditions, as varying material properties and curing conditions can cause significant differences in shrinkage [7]. In one shrinkage measurement campaign, [38] found coefficient of variations (COV) of 34% from specimens from a single batch of concrete. Thus, concretes with different mix designs and drying conditions will see even larger differences from comparisons with literature. Additionally, since the shrinkage strains were tared at day two, thermal tensile strains in the centre of the concrete may have existed at that time, resulting in additional compressive strains as the concrete cooled in addition to shrinkage strains.” The authors also compared results with model predictions (Table 4): “Table 4 shows the shrinkage strain results for the prisms, which were similar in magnitude, however, SCM1, SCM2, and C1 had unrestrained shrinkage values that were 41%, 43%, and 39% greater than the ACI prediction. This difference is relatively typical for typical shrinkage measurements compared to models [42]. … The percentage difference between the experimental data and the ACI model is even lower for many of the restrained specimens. The restrained ACI model prediction was calculated by determining the unrestrained shrinkage (considering varying volume to surface area ratios and demoulding times), and then translated this to the restrained shrinkage strain through a compatibility and equilibrium analysis of the section. Table 4 shows the differences in restrained shrinkage for the beams and slabs compared to the ACI model, which were 44% smaller for C-SCM1-F, 0.5% smaller for C-SCM2-F, 18% larger for C-C1-S, and 111% larger for NSC. The results show that the prediction from the ACI model, despite having limited available input parameters, fared relatively well for some specimens. However, for NSC, the measured results varied significantly from the prediction, and even among the other specimens, there was significant variability in the percentage differences. It is well known that the shrinkage behaviour of concrete can be hard to predict, and these results suggest that simplified models may not be the most reliable approach to estimate shrinkage.” Lastly, the results were compared with DFOS restrained shrinkage results from the limited literature dealing with DFOS measuring shrinkage ([3],[24]): “Additionally, since the shrinkage strains were tared at day two, thermal tensile strains in the centre of the concrete may have existed at that time, resulting in compressive strains due to concrete cooling in addition to shrinkage strains. These thermal strains were also found by Bado et al. [3] with DFOS.”; “Davis et al. [24] also noted sharp changes in shrinkage strain measurements in prismatic reinforced concrete specimens, however, the strains only varied from -400 micrsostrain to +50 micrsostrain near the ends of the prism, compared to a -292 microstrain average in the centre regions.”.

Round 2

Reviewer 1 Report

The authors have made the changes as per the suggestions. Only the figures should be more clear and it should be strictly technical. For example Figure 11 for the measurement the measuring tape is directly use. Instead the scale or the image of scale can be used. It should create the interest of the reader. 

Author Response

The authors have changed Figure 11 such that a scale is used rather than a measuring tape.